# Analysis of the Rice Raffinose Synthase (OsRS) Gene Family and Haplotype Diversity

**DOI:** 10.3390/ijms25189815

**Published:** 2024-09-11

**Authors:** Jinguo Zhang, Dezhuang Meng, Jianfeng Li, Yaling Bao, Peng Yu, Guohui Dou, Jinmeng Guo, Chenghang Tang, Jiaqi Lv, Xinchen Wang, Xingmeng Wang, Fengcai Wu, Yingyao Shi

**Affiliations:** College of Agronomy, Anhui Agricultural University, Hefei 230036, China; zhangjinguo@stu.ahau.edu.cn (J.Z.); 22721677@stu.ahau.edu.cn (D.M.); 22720528@stu.ahau.edu.cn (J.L.); baoyu@stu.ahau.edu.cn (Y.B.); 23720040@stu.ahau.edu.cn (P.Y.); douguohui@stu.ahau.edu.cn (G.D.); guo@stu.ahau.edu.cn (J.G.); chsoup@stu.ahau.edu.cn (C.T.); 23721081@stu.ahau.edu.cn (J.L.); 22721752@stu.ahau.edu.cn (X.W.); wangxingmeng@stu.ahau.edu.cn (X.W.); 22720535@stu.ahau.edu.cn (F.W.)

**Keywords:** rice, raffinose synthase, gene–CDS–haplotype, genetic diversity, abiotic stress, agronomic traits

## Abstract

Based on the genome information of rice (Nipponbare), this study screened and identified six raffinose synthase (RS) genes and analyzed their physical and chemical properties, phylogenetic relationship, conserved domains, promoter cis-acting elements, and the function and genetic diversity of the gene-CDS-haplotype (gcHap). The results showed that these genes play key roles in abiotic stress response, such as *OsRS5*, whose expression in leaves changed significantly under high salt, drought, ABA, and MeJA treatments. In addition, the OsRS genes showed significant genetic variations in different rice populations. The main gcHaps of most OsRS loci had significant effects on key agronomic traits, and the frequency of these alleles varied significantly among different rice populations and subspecies. These findings provide direction for studying the RS gene family in other crops.

## 1. Introduction

Raffinose synthase (RS) is a key enzyme responsible for catalyzing the transfer of galactositol to sucrose to generate raffinose residues, which is an important part of the synthetic pathway of the raffinose family of oligosaccharides (RFOs) [1]. At present, the RS gene family has been identified in *Arabidopsis thaliana* [2], *Solanum tuberosum* [3], *Triticum aestivum* [4], *Camellia sinensis* [5], *Actinidia chinensis Planch* [6], and some other plants. Some studies have shown that overexpression of *AtRS5* can enhance the tolerance and survival ability of seeds under dehydration conditions [7]. *AhRS14* and *AhRS6* play an important regulatory role in the accumulation of RFOs in peanut seeds [8]. In *Zea mays*, RS enhances the drought tolerance of plants by synthesizing more raffinose or by hydrolyzing galactose to produce more inositol [9]. In addition, *ZmDREB1A* can also regulate the expression of RS in *Zea mays*, thereby enhancing the biosynthesis of raffinose and improving the cold tolerance of plants [10]. In *Cucumis sativus*, the expression of RS increases under cold and ABA induction, thus leading to the accumulation of RFOs to resist adverse conditions [11]. In *A. thaliana*, *RS5* (At5g40390) participates in the accumulation of raffinose under abiotic stress [2]. Gil et al. found that *CmGAS2* gene expression and GAS activity are upregulated in the leaves of *Cucumis melo* L. plants under CMV infection and heat stress [12]. In wheat, the *TaRS15-3B* gene can enhance the tolerance of transgenic plants to drought and salt stress, indicating that it has a positive effect on improving the tolerance of plants to stresses [4]. These findings suggest that RS plays an important role in plant response to biotic and abiotic stresses. In addition, RS is responsible for catalyzing the synthesis of raffinose, leading to the synthesis of RFOs, which play important roles in plant stress response, signaling, photosynthesis, and tissue-specific expression [13].

Rice (*Oryza sativa* L.) is not only an important food crop, but it is also a valuable resource for scientific research and breeding improvement due to its rich genetic diversity [14]. However, it has been estimated that by 2035, the world will need an additional amount of 116 million tons of rice to meet the needs of the growing population. This goal must be achieved by taking into account a variety of biotic and abiotic factors affecting rice production, especially extreme weather [15]. Wang et al. have completed the sequencing of 3010 rice materials (3KRG), which reveals the rich genetic variation and diversity of rice and provides valuable genetic resources for future rice breeding and sustainable agriculture [16]. Zhang et al. characterized the gcHap (gene–CDS–haplotype) diversity of 45,963 genes from 3010 rice varieties. The constructed comprehensive gcHap dataset shows advantages over SNPs in identifying causal genes for complex traits, providing an important resource and direction for future rice genetic improvement and breeding [17]. Moreover, Zeng et al. combined the haplotype diversity analysis of 3010 rice genomes to study the GLR gene family, revealing its application potential in rice yield enhancement and stress resistance improvement [18]. In a similar way, Cheng et al. found that *OSBES1-4* play a key role in regulating rice grain size [19].

With the development of molecular biology and genomics technology, there have been many studies of RS genes in terms of molecular structure, expression pattern, and functional roles. However, there have been few studies focusing on the number, evolutionary relationship, and specific functions of RS gene family members in rice, and there has also been limited research on the genetic diversity, allelic variation, and relevant agronomic traits of RS genes in different rice populations.

In order to better explore the functions of rice genes to achieve the goal of increasing rice yield, future rice improvement must be based on innovative breeding techniques and strategies, which requires a more complete understanding of the function and diversity of all rice genes and gene networks and their associations with important agronomic traits of rice.

In this study, a comprehensive approach was adopted to systematically identify and characterize the rice RS gene family members, analyze their expression patterns in different tissues, explore their response mechanism to abiotic stress, and evaluate their genetic diversity and allelic variation in different rice populations. By identifying RS candidate genes from the rice genome, the gene structure, phylogeny, and expression pattern were studied, which provides a basis for revealing the function of the RS gene family. In addition, a comprehensive population genetic analysis on the gcHap diversity of OsRS genes was carried out to provide theoretical and practical support for genetic improvement and molecular design breeding of rice.

## 2. Results

### 2.1. Genome-Wide Identification and Characterization of OsRS Genes

We successfully identified the OsRS gene family members through BLAST and HMM retrieval of the whole Nipponbare genome. After final screening, six OsRS genes were identified in the genome. In order to further understand the characteristics of these genes, we analyzed their physical and chemical properties, including protein length, molecular weight (Da), isoelectric point (PI), instability index, aliphatic index, hydrophilic index, and subcellular localization (Appendix A). The sequence length of these OsRS proteins ranged from 338 to 925 amino acids (aa), and the molecular weight ranged from 36,614.52 Da (*OsRS1*) to 99,214.8 Da (*OsRS4*), with an average of about 72,064.3 Da. The PI ranged from 5.38 to 8.36, with an average of 6.06. Notably, 66.67% of the OsRS proteins (4/6) had an instability index lower than 40, indicating that most of the OsRS proteins are stable. Moreover, the minimum aliphatic index was 77.37, indicating that OsRS proteins have certain degrees of thermal stability. All OsRS proteins had a negative hydrophilic index, indicating that they are hydrophilic. Subcellular localization analysis showed the presence of OsRS proteins in a variety of organelles, suggesting that they may have a variety of biological functions. These results revealed that the members of the OsRS protein family have different physicochemical properties, indicating that they may play different biological roles in rice.

In terms of chromosomal distribution, the six OsRS genes were located on chromosomes 1, 3, 4, 6, 7, and 8 of Nipponbare, respectively, while none of them was found on other chromosomes (Figure 1a). To further explore the evolutionary and structural characteristics of OsRS genes, we constructed a phylogenetic tree containing RS genes of *Oryza sativa* L. (Nipponbare), *Glycine max*, *Arabidopsis thaliana*, *Setaria viridis*, *Sorghum bicolor*, and *Cucumis sativus* (Appendix A). These genes were divided into three major subfamilies (I–III) based on their evolutionary proximity (Figure 1b). This classification can help understand the evolution of RS genes in different species and their possible structural and functional differences.

Analysis and prediction on the MEME website (https://meme-suite.org/meme/, accessed on 26 November 2023) were carried out to investigate the conserved motifs in the OsRS protein family (Figure 1c). The results showed that the protein motifs within the OsRS subfamilies were highly similar in both type and quantity. However, when different subfamilies were compared, these motifs showed significant differences. In particular, motif 1 was prevalent in all members of the OsRS family, suggesting that it is essential for the structure and function of OsRS proteins. In addition, *OsRS3*, *OsRS4*, *OsRS5*, and *OsRS6* all contained motifs 1 to 10, suggesting that they may perform similar biological functions.

An analysis of the conserved domains of proteins revealed that all OsRS proteins contain an AmyAc_family superfamily domain (Figure 1c), which is consistent with the analysis results of conserved motif. Moreover, different subfamilies of OsRS genes also showed great differences in gene length and exon number, and these differences were generally consistent with the analysis results of conserved motifs and conserved domains. These findings provide insights into the structural and functional diversity of the OsRS protein family members and may help further explore their roles in biological processes.

In the analysis of promoter regions of six OsRS genes, a total of 38 different cis-acting elements were identified (Figure 2a). Among these elements, TATA-box and CAAT-box, which are closely related to transcriptional initiation, accounted for the largest proportion, indicating that the OsRS genes have normal transcriptional activity. Further analysis of these cis-acting elements revealed that about 20% of them (554/2782) are associated with stress response. Among these stress responsive elements, those related to the response to methyl jasmonate (MeJA) and abscisic acid (ABA) were particularly prominent, accounting for 41.85% and 33.70% of the hormone-responsive elements, respectively. In addition, the elements related to light response accounted for about 14% (371/2782), while those related to plant growth were relatively fewer, accounting for only 5.9% (163/2782) of the total. It is particularly noteworthy that elements related to drought and salt stress dominated the environmental stress response elements, accounting for 80.87% of the total.

As shown in Figure 2b, heat map analysis of cis-acting elements in the promoter region of OsRS genes revealed that MYB, MYC, ABRE, CGTCA-motif, TGACG-motif, and other elements related to environmental response and hormonal response were present in every member of the OsRS gene family, suggesting that the OsRS gene family may play an important role in plant response to stress and hormonal stress through these cis-acting elements.

### 2.2. Collinearity Analysis of RS Genes

In order to further explore the evolutionary relationship of RS family genes in different species, we used the MCScanX toolkit to analyze the phylogenetic relationship of RS family genes. The analysis results revealed that rice (Nipponbare) had five pairs of collinear RS genes with *Sorghum bicolor (Sb)*, four pairs of collinear RS genes with *Setaria viridis (Sv)*, three pairs of collinear RS genes with *Glycine max (Gm)*, and one collinear RS gene pair with *Cucumis sativus (Cs)* (Figure 3). These findings suggested that the OsRS genes have more collinear genes with monocotyledonous plants sorghum and setaria than with dicotyledonous plants, suggesting conservation of RS proteins during species evolution. This comparative analysis of different species could help better understand the evolutionary dynamics of RS family genes in different plants and their possible common functions in plant biology.

### 2.3. Ka/Ks Ratio Analysis of OsRS Homologous Genes

The results of collinear analysis showed that the sequence similarity of *OsRS2* and *OsRS5* reached more than 80% (Figure 4), indicating the homology between them. In addition, it was found that these gene pairs were paralogs that evolved through gene replication events. In order to further explore the influence of selection pressure on the evolution of the OsRS family, we calculated the non-synonymous replacement rate (Ka), synonymous replacement rate (Ks), and the Ka/Ks ratio of OsRS family homologous genes. The analysis results showed that the Ka/Ks ratio of a pair of homologous genes in the OsRS family was less than 1 (Table 1), indicating that during the evolution of the OsRS family, these genes were subjected to strong purification selection pressure [20].

### 2.4. Expression Profile of OsRS Genes in Different Tissues and under Abiotic Stress

The expression of the OsRS gene family showed obvious tissue specificity in rice. For example, the expression levels of *OsRS3*, *OsRS4*, and *OsRS5* were higher in flower tissues. In leaves, *OsRS1*, *OsRS4*, and *OsRS5* showed high expression. In the root, *OsRS3*, *OsRS4*, and *OsRS6* were expressed at high levels, while in the stem, *OsRS6* was highly expressed. The expression levels of OsRS genes in different tissues are shown in Figure 5a. The expression pattern of OsRS genes also changed under stress conditions. Under drought stress, the expression of *OsRS5* in the shoot significantly increased after 6, 12, and 24 h, while that of *OsRS2* in the root and shoot showed almost no change (Figure 5b). Under high salt stress, the expression of *OsRS5* and *OsRS1* increased in the stem and leaf (Figure 5c). In addition, after ABA treatment, the expression levels of *OsRS4* and *OsRS5* in the root and shoot also increased (Figure 5d). After MeJA treatment, the expression of *OsRS4* in the root and shoot also increased (Figure 5e). Notably, *OsRS2* was almost not expressed after ABA and MeJA treatment. The expression of OsRS genes also showed certain specificity under temperature stress. After heat treatment, the expression levels of *OsRS4* and *OsRS5* in the stem and leaf increased (Figure 5f). Under cold stress, the expression levels of *OsRS3*, *OsRS4*, and *OsRS6* in the root increased, and those of *OsRS1* and *OsRS5* in the buds also increased (Figure 5g). These results revealed the expression patterns of OsRS genes in different tissues and under environmental stresses, providing important clues for further studying the functions of these genes in rice growth and development and stress response.

### 2.5. Real-Time Fluorescence Quantitative PCR Analysis of OsRS Genes

Based on the comprehensive cis-acting element analysis and RNA-seq data, we selected five genes (*OsRS1*, *OsRS3*, *OsRS4*, *OsRS5*, and *OsRS6)* as subjects and analyzed their expression patterns under different conditions (Figure 6). Except for *OsRS1*, all other four genes showed significant changes in expression under ABA treatment. In particular, the expression of *OsRS3*, *OsRS4*, and *OsRS6* peaked after 4 h, and that of *OsRS5* reached the highest level at 24 h. MeJA treatment resulted in similar results to ABA treatment, with other genes except for *OsRS1* showing significant upregulation of expression. Specifically, the expression levels of *OsRS4* and *OsRS6* were significantly upregulated and peaked at 12 h and 8 h, respectively. Under drought stress treatment, the expression of *OsRS1* gradually increased. The expression level of *OsRS4* and *OsRS5* increased dramatically at 8 h, reaching the highest value followed by decreases. Under heat stress, only the expression of *OsRS1* increased significantly and peaked at 24 h. After cold stress treatment, the expression of all four genes reached the peak at 24 h except for *OsRS4* (at 12 h). The expression of all genes changed significantly under NaCl stress, with the most significant increase being observed in *OsRS5*. In particular, the expression of all *OsRS4*, *OsRS5*, and *OsRS6* peaked at 4 h, while that of *OsRS1* and *OsRS3* reached the maximum at 12 h and 24 h, respectively. These results indicated that these genes have specific expression patterns in response to different environmental signals and hormone treatments, which may be related to their different functions in rice growth and stress response. The specific mechanisms of action of these genes need to be further investigated.

### 2.6. Genetic Diversity and Allelic Diversity of OsRS Loci in Rice Population

Based on the coding sequence (CDS) haplotype data (gcHap), we calculated the Shannon fairness (*E_H_*) value, the frequency of gcHaps, and the number of major gcHaps (gcHaN) of the six OsRS genes in four major rice populations (with frequency ≥1% in 3KRG) (Appendix A). Across the six OsRS genes, the mean gcHaps, the major gcHaps, and *E_H_* values were 94, 7.3, and 0.229, respectively, indicating their differences in genetic diversity. In particular, *OsRS5* had the highest *E_H_* value of 0.34, while *OsRS6* had the lowest *E_H_* value of 0.129. This difference corresponded to their respective numbers of gcHaps, with *OsRS5* having 164 gcHaps and *OsRS6* only having 10 gcHaps (Appendix A).

Upon further observation of the genetic diversity among different rice populations, we found that the average *E_H_* values of the six OsRS genes also showed significant differences among populations. The average *E_H_* was 0.212, 0.154, 0.26, 0.288, and 0.46 (the highest) for the *Xian*, *Geng*, Aus, Bas, and admix populations, respectively. Among these populations, the average number of detected gcHaps and the main gcHaps also varied, which were 65.5 and 6.5 for the *Xian* population, 28.8 and 4.7 for the *Geng* population, 17.5 and 6.2 for the Aus population, 11.3 and 4.5 for the Bas population, and 23.7 and 7.8 for the admix population, respectively (Appendix A).

To understand the genetic differences of OsRS genes among major rice populations, we pairwise analyzed the gcHap data of six OsRS genes between populations using the genetic diversity index (*I_Nei_*). The results showed that *OsRS1*, *OsRS4*, and *OsRS5* exhibited significant genetic differentiation between Aus-XI and Aus-GJ populations (*I_Nei_* < 0.35) (Figure 7, Appendix A). In particular, *OsRS5* also showed strong genetic differentiation among Aus-XI, Aus-GJ, Aus-Bas, XI-GJ, and XI-Bas populations. These results suggested that allelic variation of OsRS gene loci significantly contributes to the differentiation of rice populations and their adaptability to different environments.

### 2.7. Effects of Modern Breeding on gcHap Diversity of OsRS Genes

In order to explore the effects of modern breeding on the gcHap diversity of OsRS genes in recent decades, we compared the genetic diversity of OsRS genes in modern varieties (MVs) and local varieties (LANs). The study included 732 local (LANs—*Xian*) and 358 modern (MVs—*Xian*) varieties of *Xian*, and 328 local (LANs—*Geng*) and 139 modern (MVs—*Geng*) varieties of *Geng* (Table 2 and Table 3). In the *Xian* population, the average *E_H_* value of the six OsRS genes in MVs—*Xian* was 0.329, which is 0.079 higher than that of LANs—*Xian*. The data showed that almost all OsRS genes in MVs—*Xian* had higher genetic diversity than LANs—*Xian*. However, it is worth noting that despite the higher genetic diversity, MVs—*Xian* had 7.3 fewer gcHaps/locus on average than LANs—*Xian*.

In the Xian population, MVs—*Xian* gained an average of 20.3 new gcHaps/locus, which were not present in LANs—*Xian*, suggesting that new gcHaps were generated by intragenic recombination during breeding. The increase in genetic diversity and decrease in major gcHaN observed at multiple RS loci reflect significant changes in the frequency of major gcHaps at these loci during breeding. In fact, except for *OsRS2*, the major gcHap frequency F_(P)_ of the other five RS loci showed significant changes, including significant decreases at four loci and significant increases at one locus (Table 2).

The average *E_H_* of the six OsRS genes in the *Geng* population was 0.21 in MVs—*Geng* and 0.171 in LANs—*Geng*. In particular, the genetic diversity of modern varieties showed a significant decline only at the *OsRS4* locus. The average value of gcHaps/locus for MVs—*Geng* was 12, and that for LANs—*Geng* was 18 (Table 3). Of the six OsRS loci, only *OsRS6* showed a significant decrease in F_(P)_. In addition, the same dominant gcHap Hap1 was observed at all OsRS loci, suggesting that Hap1 plays an important role in both *Xian* and *Geng* populations (Appendix A).

### 2.8. Comparison of Trait Values between the Favorable and Unfavorable gcHaps of OsRS Genes in Rice

In rice populations, the dominant gcHap (the most frequent haplotype) in an RS locus is generally believed to have been favored by natural selection during evolution. Conversely, those major gcHaps with the lowest frequency in the population may be considered unfavorable haplotypes [18]. We compared and analyzed the difference between favorable and unfavorable gcHaps on 15 agronomic traits in the six OsRS loci. Among the 90 comparisons, 35 (38.8%) comparisons showed significant phenotypic differences (Appendix A). Among the six OsRS genes, some specific gcHaps of four genes showed significant phenotypic differences in 1000-grain weight (TGW) and days to heading (DTH). Specifically, among the 15 traits examined, *OsRS1* showed the most phenotypic differences between favorable and unfavorable gcHaps, with a total of 11 traits. In sharp contrast, no phenotypic differences were observed between the two gcHap variants of the *OsRS3* gene, suggesting that other OsRS genes may affect more traits than *OsRS3* (Appendix A).

### 2.9. Correlation Analysis of Major gcHaps of OsRS Genes and Important Agronomic Traits

To verify the functional importance of the six OsRS genes in rice, we constructed a gcHap network for the dominant alleles of these genes in five rice populations, and correlated these alleles in 3KRG materials with four agronomic traits, including panicle number per plant (PN), panicle length (PL), plant height (PH), and TGW (Figure 8 and Appendix A). Out of a total of 24 combinations (6 genes × 4 traits) analyzed, 16 (65.9%) showed very significant associations (*p* < 10^−7^), indicating that the major alleles of many OsRS genes are strongly associated with phenotypic values for one or more traits.

An in-depth analysis of OsRS genes revealed that *OsRS2* and *OsRS6* are conserved genes, with each having four major gcHaps. Specifically, Hap3 of *OsRS2* had the highest frequency in LANs—*Xian*, while Hap2 had the highest frequency in LANs—*Geng*. Similarly, Hap4 of *OsRS6* had the highest frequency in LANs—*Xian*, while Hap2 had the highest frequency in LANs—*Geng* (Figure 8 and Appendix A). In *OsRS2*, Hap3 had a non—synonymous mutation relative to Hap2, and compared with Hap2, Hap3 significantly improved the value of PN during breeding. Further analysis revealed that *OsRS4* had three major gcHaps, with six non-synonymous mutations occurring between Hap2 and Hap3. Hap3 was the most frequent haplotype in both LANs—*Xian* and LANs—*Geng*. However, during breeding, Hap2 showed no significant change in PN, PL, PH, and TGW traits compared with Hap3. These results suggested that some specific OsRS gene alleles may have significant effects on some important agronomic traits of rice.

### 2.10. Mining of Favorable Alleles at OsRS Loci for Yield Improvement

In rice, the major gcHaps at most OsRS loci showed significantly different effects on key agronomic traits. The frequency of favorable alleles varied greatly for different yield traits, OsRS loci, and rice populations. For example, Hap4 of *OsRS6* showed the highest correlation with TGW in 3KRG germplasm, with a fixed frequency in MVs—*Xian*, but a relatively low frequency in MVs—*Geng*. In addition, Hap2 of *OsRS1* had the highest correlation with GL, TGW, GW, and PL in 3KRG germplasm. Hap2 of *OsRS2* had the highest correlation with TGW and GW. Hap4 of *OsRS3* had the highest correlation with CN. However, Hap4 of *OsRS5* had the highest correlation with PL (Figure 9). These results suggested that for different yield traits, the favorable alleles may differ significantly between two rice subspecies or in different genetic backgrounds, and these differences may also be influenced by the environmental conditions. This means that in rice breeding, the allelic diversity of different loci and their responses to specific environmental conditions need to be taken into account for more efficient use of genetic resources and breeding of adaptable and high-yield rice varieties.

## 3. Discussion

In this study, based on rice (Nipponbare) genome information, a total of six OsRS genes were identified, all of which had the AmyAc_family superfamily domain and were divided into three subfamilies. Through chromosomal localization analysis, these genes were found to be distributed on chromosomes 1, 3, 4, 6, 7, and 8 of rice. Phylogenetic analysis of different species revealed the evolutionary relationship between OsRS genes and RS genes in other species, indicating their conservation and possible functional differences in different species.

We analyzed the expression patterns of OsRS genes in different tissues and found that they were specifically expressed in flower tissues, leaves, roots, and stems, suggesting their potential role in the growth and development of rice. An analysis of the RNA-seq data of the OsRS gene family (Figure 5) and the qPCR results (Figure 6) showed that different OsRS genes had specific expression responses to environmental signals and stress conditions such as ABA, MeJA, drought, heat, cold, and salt stress. This temporal and spatial expression specificity suggests that they may have different functions in rice growth and development and stress response. *OsRS1* showed regulatory effects under multiple stress conditions, while *OsRS5* exhibited unique expression patterns under ABA and MeJA treatments. The expression of *OsRS4* and *OsRS6* was upregulated under hormone treatment and was rapidly responsive to salt stress. The expression patterns of these genes are closely related to their biological functions, but the specific mechanism of action still needs further study.

Based on the gcHap data of the 3KRG project, the OsRS gene family was found to have significant genetic diversity in different populations, among which the admix population has the highest genetic diversity. Genetic differentiation analysis showed that *OsRS1*, *OsRS4*, and *OsRS5* were significantly different among different rice populations, with important impacts on the adaptation and population differentiation of rice. A comparison of modern and local varieties showed that although the modern varieties in the *Xian* population have high genetic diversity, they have a small number of gcHaps, which is possibly a result of gene recombination. In the *Geng* population, the modern varieties have lower genetic diversity at the *OsRS4* locus, but the major gcHap Hap1 has a consistent frequency at all loci, indicating its stability across different populations. These results indicate that breeding has a profound effect on the genetic diversity of rice, and allelic diversity and environmental adaptability should be considered in breeding.

By comparing the performance of favorable and unfavorable gcHaps on 15 agronomic traits in six OsRS loci, it was found that *OsRS1* showed the most phenotypic differences between favorable and unfavorable gcHaps, suggesting that this gene has significant effects on several agronomic traits. In addition to that of *OsRS3*, some specific gcHaps of other OsRS genes were significantly associated with traits such as TGW and DTH, highlighting the need to consider allelic diversity in rice breeding. An in-depth association analysis showed that the gcHaps of major alleles of six OsRS genes were significantly correlated with agronomic traits such as PN, PL, PH, and TGW in five rice populations, and very significant associations were observed in 65.9% of the analyzed cases. In particular, the frequencies of major gcHaps and non-synonymous mutations in conserved genes *OsRS2* and *OsRS6* in different populations were significantly correlated with phenotypic values of specific agronomic traits, providing potential genetic markers for rice genetic improvement.

Our results highlighted that the major gcHaps of OsRS loci have a significant influence on key agronomic traits, and the frequency of these alleles varies significantly across rice populations and subspecies. For example, Hap4 of *OsRS6* is highly correlated with TGW, but its frequency varies across populations and may be influenced by genetic background and environmental factors. Other gcHaps of *OsRS1*, *OsRS2*, *OsRS3*, and *OsRS5* also showed high correlations with different agronomic traits. These results highlight the necessity to consider allelic diversity and environmental adaptability in the breeding of rice, and select and utilize favorable alleles to improve rice adaptability and yield.

## 4. Materials and Methods

### 4.1. Identification of the OsRS Gene Family Members

The rice (Nipponbare) genome sequence and annotation information were first obtained from the Ensembl the Plants database (https://plants.ensembl.org/index.html, accessed on 25 November 2023) [21]. Next, the Pfam database (http://pfam.xfam.org/, accessed on 26 November 2023)was accessed, and the Hidden Markov Model (HMM) of PF05691 was downloaded [4]. By using the Simple Hmm Hearch tool of TBtoolsV2.118 software, the gene and protein sequences that may belong to the OsRS gene family were screened and identified [22]. From the NCBI web site (https://www.ncbi.nlm.nih.gov/Structure/bwrpsb/bwrpsb.cgi, accessed on 26 November 2023), by using the domain analysis tools, the domains of these genes were analyzed in detail, and the RS gene family members were screened and identified by combining the functional annotation for comprehensive evaluation. These genes were renamed based on their location on the chromosome. In addition, the Cell-PLoc 2.0 (http://www.csbio.sjtu.edu.cn/bioinf/euk-multi-2/, accessed on 27 November 2023), online tools were used for rice subcellular localization of OsRS genes [23]. Finally, through the Protein Paramter Calc function of TBtools software, protein physico-chemical properties of rice RS genes, such as isoelectric point (PI) and molecular weight (Da), were determined [22].

### 4.2. Phylogenetic Analysis of OsRS Genes

To study the phylogenetic relationship of RS genes in rice (Nipponbare), *Arabidopsis thaliana*, *Cucumis sativus*, *Glycine max*, *Sorghum bicolor*, and *Setaria viridis*, we obtained the RS genes from these plant species and constructed a phylogenetic tree by utilizing the Neighbor—Joining method in MEGA11 software. In order to display the phylogenetic tree more intuitively, we adopted the online tool iTOL (Interactive Tree Of Life) to beautify it [24].

### 4.3. Analysis of Cis-Acting Elements, Conserved Motifs, Conserved Domains, and Promoter Regions of OsRS Family Proteins

The required gene structure annotation files were downloaded from the Ensembl Plants website, and TBtools was used to visualize the data. Then, the identified OsRS protein sequences were uploaded to the MEME website (https://meme-suite.org/meme/tools/meme, accessed on 26 November 2023), and the online tool was used to identify the sequences of conserved motifs [25]. In addition, the conserved domains of OsRS proteins were queried through the NCBI Protein Batch CD-search database [26].

In order to further analyze cis-acting elements of OsRS gene family members, the Gtf/Gff3 sequence extraction tool of TBtoolsV2.118 software was used to extract promoter sequences from the upstream 2000 bp region of CDS. Subsequently, these sequence data were submitted to the PlantCARE database for a detailed cis-acting element analysis [27]. Finally, the Simple BioSequence Viewer function of TBtoolsV2.118 software was used again to visually display the obtained cis-acting components [22].

### 4.4. Collinear Analysis of OsRS Genes

The required gene structure annotation file was downloaded from the Ensembl Plants website, and the interspecific and intraspecic collinearity analysis maps were drawn using TBtools’ One Step MCScanX and Advance Circos functions.

### 4.5. Gene Expression Profile Based on RNA-seq

In this study, we obtained the expression data of OsRS genes in different tissues and developmental stages and under different stresses from the RNA-seq database (http://ipf.sustech.edu.cn/pub/plantrna/, accessed on 28 November 2023) [28]. TBtools and Adobe Illustrator 2023 software were used for data visualization.

### 4.6. Material Handling

The rice seeds (the Nipponbare preserved in our laboratory) were first disinfected with 3% sodium hypochlorite for 30 min and then germinated at 28 °C for 3d. After germination, rice seedlings with similar growth profiles were selected and transplanted into hydroponic boxes containing the Hoagland nutrient solution in the intelligent light temperature incubator (under normal conditions: 28 °C/12 h in the day, 26 °C/12 h at night, humidity 80%, light intensity 3000 lux).

When the seedlings reached the three-leaf stage, they were subjected to a series of stress treatments, including cold (4 °C), heat (42 °C), high salt (200 mmol/L NaCl), drought simulation (20% PEG6000), and two different hormones (100 μmol/L ABA, 100 μmol/L MeJA). Rice leaf samples were taken at 0, 4, 8, 12, and 24 h, respectively. After collection, the samples were quickly frozen in liquid nitrogen and then transferred to −80 °C for storage to extract the total RNA for gene expression analysis.

### 4.7. Real-Time Fluorescence Quantitative PCR Analysis

The obtained sample was ground in liquid nitrogen with a mortar and pestle, and the TaKaRa MiniBEST Plant RNA Extraction Kit (TaKaRa, Beijing, China) was used to extract the RNA, which was then reverse transcribed by the TaKaRa Reverse Transcription Kit (TAKaRa, Beijing, China). Primers were designed for the OsRS gene family, and relevant primer information is shown in Appendix A. Loc-Os03g61971.1 was used as the internal reference to conduct real-time fluorescence quantitative detection by the LightCycler 96 quantitative PCR instrument (Roche Diagnostics GmbH, Sandhofer Strasse 116, 68305 Mannheim, Germany). The experiment was set up for three biological and three technical repeats, and the relative expression levels of each gene were calculated by the 2^−ΔΔct^ method. WPS 2023 and GraphPad Prism 8 software were used to complete data statistics and chart drawings to explore the expression patterns of genes under different stresses.

### 4.8. OsRS Gene gcHaps and gcHap Diversity in Modern and Local Varieties

Shannon’s evenness (*E_H_*) index was used to assess gcHap diversity at specific OsRS loci in different rice populations. The rice OsRS gene gcHap data were downloaded from the RS RFGB website (https://www.rmbreeding.cn/Index, accessed on 18 December 2023), and the Nei’s gene’m (*I_Nei_*) of each gene was calculated to measure the genetic differences and genetic differentiation between two rice populations [17,29]. To investigate the effect of modern breeding on the OsRS gene gcHap diversity, we collected detailed information of 3010 3KRG rice materials, including 732 *Xian* (indica) local varieties (LANs—*Xian*), 358 *Geng* (japonica) local varieties (LANs—*Geng*), 328 *Xian* (indica) modern varieties (MVs—*Xian*), and 139 *Geng* (japonica) modern varieties (MVs—*Geng*). By using the R scripts and GraphPad Prism 8 software, the drift frequencies of major gcHaps in OsRS genes were calculated [18], and the comparison of gcHap distributions between modern and local varieties was visualized.

### 4.9. Extraction of Major gcHap Phenotypes for OsRS Genes

The phenotypic data of 3010 Asian cultivated rice samples for 15 traits were downloaded from RFGB (https://www.rmbreeding.cn, accessed on 18 December 2023), including days to heading (DTH, day), plant height (PH, cm), flag leaf length (FLL, cm), flag leaf width (FLW, cm), panicle number (PN, count), panicle length (PL, cm), culm number (CN, count), grain length (GL, mm), grain width (GW, mm), grain length/width ratio (GLWR, ratio), thousand grain weight (TGW, g), leaf rolling index (LRI, %), seedling height (SH, cm), and ligule length (LL, mm) [18]. The correlation between the major gcHaps of OsRS genes and agronomic traits in 3010 samples was then analyzed using an R script. The statistical significance was assessed by univariate ANOVA and a Tukey multiple comparison test. The results were arranged and embellished using Adobe Illustrator 2023.

### 4.10. Construction of CDS Haplotype (gcHap) Network of OsRS Genes

R-encapsulated pegas was first used to construct the haplotype (gcHap) of OsRS genes [30]. Subsequently, a statistical reduction algorithm was used to generate a gcHap network for each OsRS gene, which connects closely related haplotypes with a minimum of mutation steps [31]. Detailed steps refer to Zhang’s article [17] published in *Molecular Plant*. Finally, Adobe Illustrator 2023 software was used to complete the layout design of the image.

## Figures and Tables

**Figure 1 ijms-25-09815-f001:**
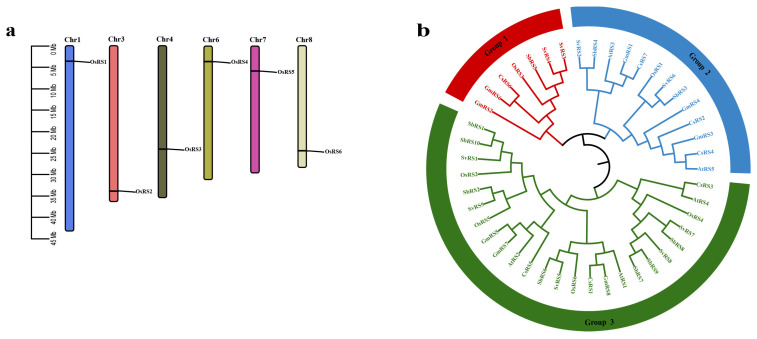
Characteristics of OsRS genes. (**a**) Chromosomal localization of OsRS genes. (**b**) Phylogenetic trees of OsRS genes from Nipponbare (Os), *Glycine max (Gm)*, *Arabidopsis thaliana (At)*, *Setaria viridis (Sv)*, *Sorghum bicolor (Sb)* and *Cucumis sativus (Cs)*. (**c**) Phylogenetic tree, motif prediction, domain, and exon-intron distribution of OsRS genes from left to right.

**Figure 2 ijms-25-09815-f002:**
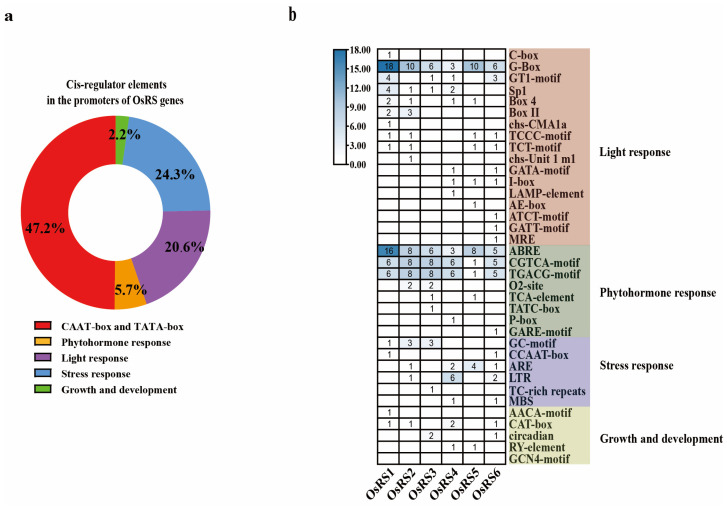
Analysis of cis-acting elements of the OsRS gene. (**a**) Distribution and proportion of cis-acting elements in the promoter region of the OsRS gene; different colors represent different proportions. (**b**) Heatmap analysis of cis-acting elements in the promoter region of OsRS genes. In the heatmap, the numerical values represent the quantity of different cis-acting elements, with darker colors indicating higher quantities.

**Figure 3 ijms-25-09815-f003:**
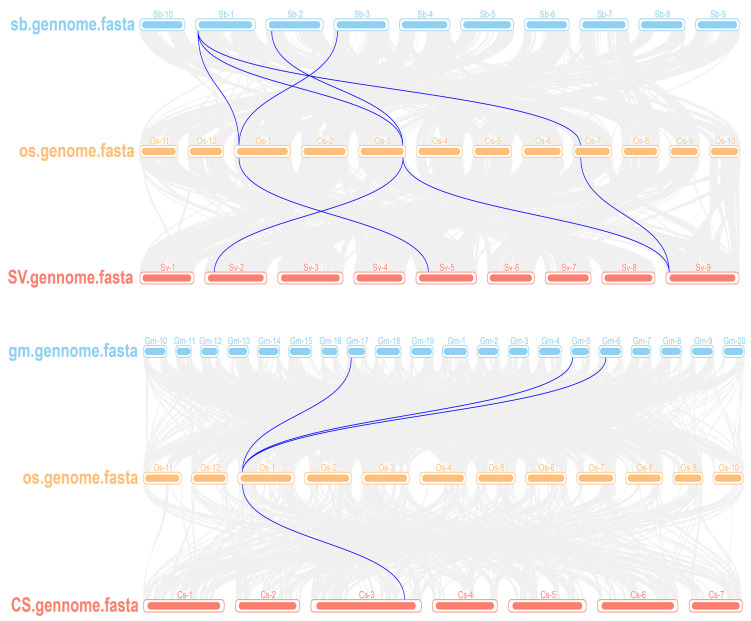
Collinear relationship between OsRS genes and genes from other species. The collinear regions of the genome of rice (Nipponbare) and other species are represented by grey lines and collinear gene pairs by blue lines. *Oryza sativa* L. (Nipponbare) is represented by *Os*; *Glycine max* is represented by *Gm*; *Sorghum bicolor* is represented by *Sb*; *Setaria viridis* is denoted by *Sv*; and *Cucumis sativus* is represented by *Cs*.

**Figure 4 ijms-25-09815-f004:**
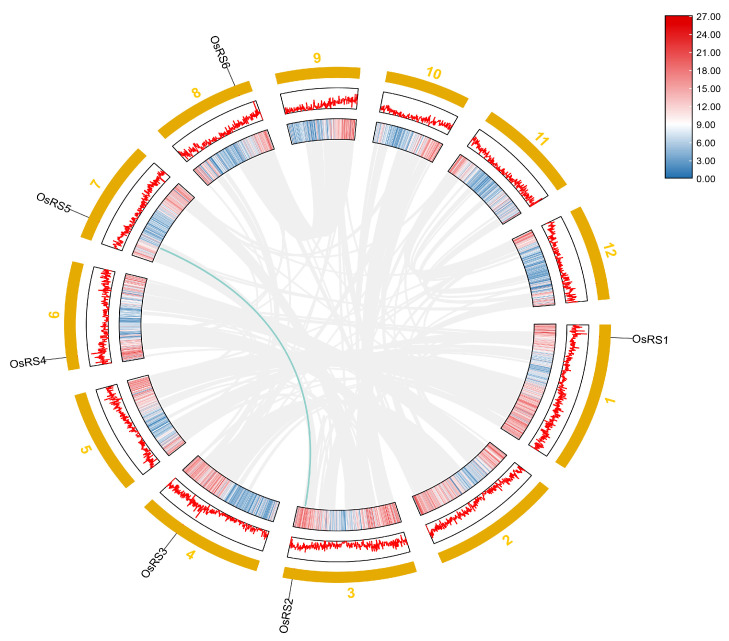
Homologous relationship and chromosomal localization of OsRS genes. The line indicates a homologous relationship.

**Figure 5 ijms-25-09815-f005:**
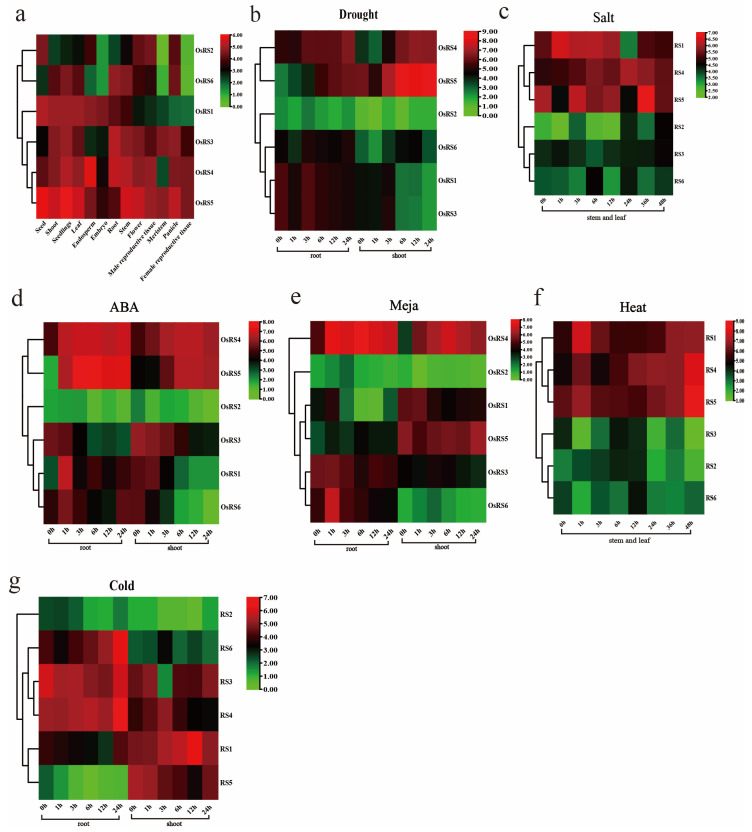
Analysis of OsRS gene expression. Color markers indicate changes in gene expression. Red indicates high expression, and green indicates low expression. (**a**) Expression of OsRS genes in the leaf, root, seedling, stem, flower, embryo, shoot, meristem, male reproductive tissue, female reproductive tissue, panicle, and seed. (**b**) Expression levels of OsRS genes in the root and shoot after drought stress. (**c**) Expression levels of OsRS genes in the stem and leaf of rice after high salt stress. (**d**) Expression levels of OsRS genes in the root and shoot after ABA hormone treatment. (**e**) Expression levels of OsRS genes in the root and shoot after MeJA treatment. (**f**) Expression levels of OsRS genes in the stem and leaf after heat treatment. (**g**) Expression levels of OsRS genes in the root and shoot after cold stress.

**Figure 6 ijms-25-09815-f006:**
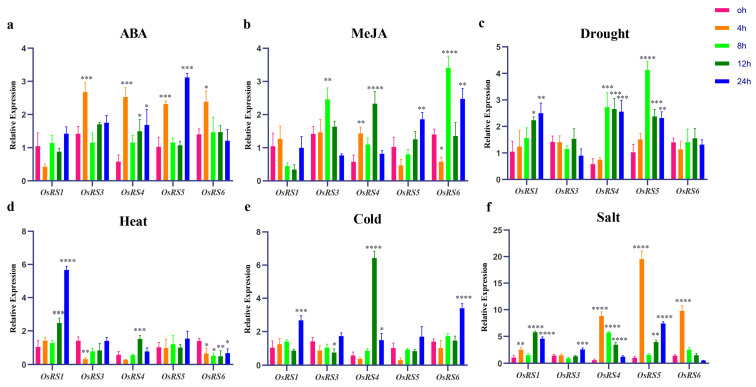
Analysis of expression levels of five genes of the OsRS family under different treatments: (**a**) 100 μmol/L ABA treatment; (**b**) 100 μmol/L MeJA treatment; (**c**) 20% PEG6000 simulated drought stress. (**d**) 200 mmol/L NaCl simulated salt stress; (**e**) 42 °C heat treatment; (**f**) 6 °C cold treatment. Statistical analysis of the data was performed using WPS2023 software, and IBM SPSS Statistics 25 statistics analysis software was used to perform analysis of variance; the significance level was defined as **** *p* < 0.0001, *** *p* < 0.001, ** *p* < 0.01, * *p* < 0.05.

**Figure 7 ijms-25-09815-f007:**
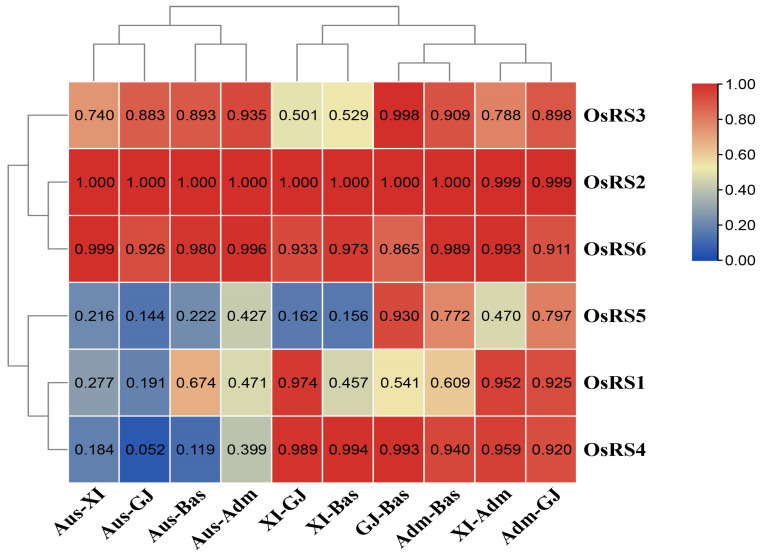
Genetic diversity index (*I_Nei_*) of OsRS genes in pairwise comparison of different populations calculated from gcHap data.

**Figure 8 ijms-25-09815-f008:**
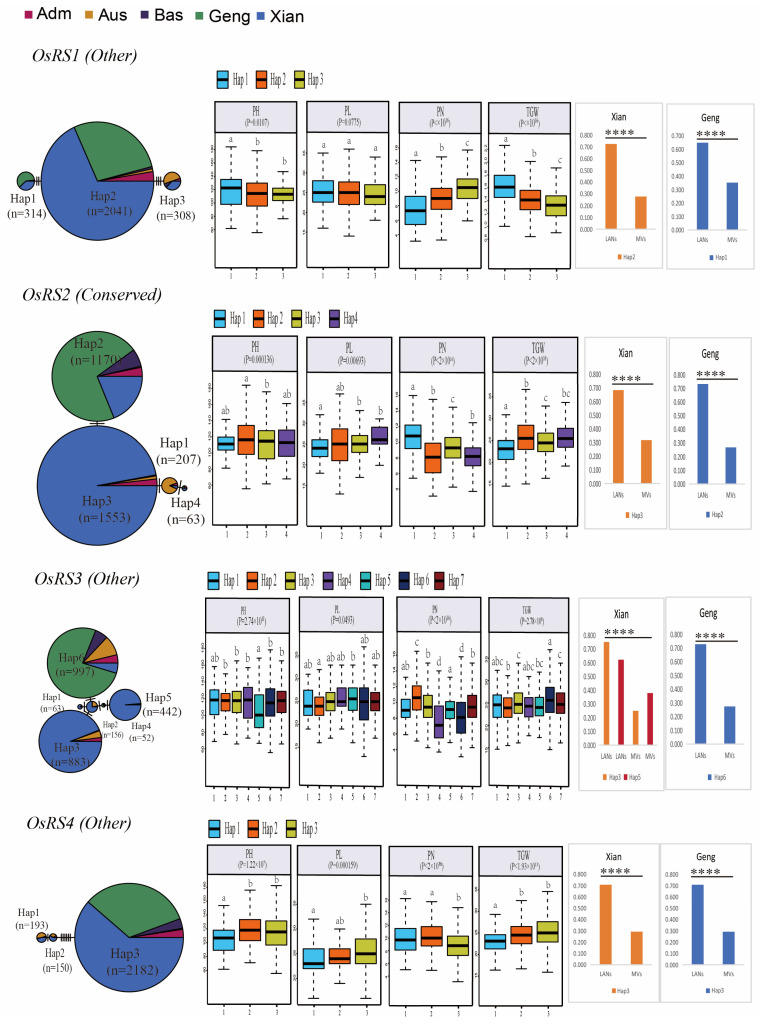
Haplotype networks of four cloned OsRS genes and four associated agronomic traits in 3KRG. The letters indicate differences between haplotypes assessed by two-factor ANOVA, where different letters on the boxplot indicate statistically significant differences at *p* < 0.05 based on Duncan’s multirange test. The bar chart on the right shows the difference in frequency of major gcHaps between local varieties (LANs) and modern varieties (MVs) of *Xian* and *Geng*. A chi-square test was used to determine significant differences in the proportion of a gcHap between different populations **** *p* < 0.0001.

**Figure 9 ijms-25-09815-f009:**
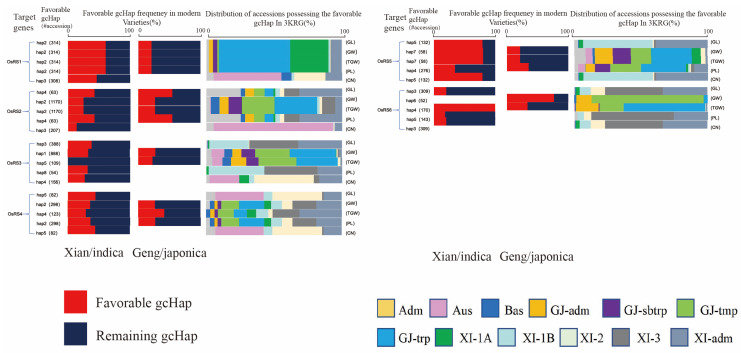
Favorable gcHap frequencies of six OsRS genes affecting TGW, GL, GW, PL, and CN in Xian/indica (XI), Geng/japonica rice (GJ), and different rice subpopulations.”#accession”indicates the number of accessions that possess the favorable gcHap.Five subpopulations of XI (XI—1A, XI—1B, XI—2, XI—3, and XI—adm) and four subpopulations of GJ (temperate GJ: GJ—tmp, subtropical GJ: GJ—sbtrp, tropical GJ: GJ—trp, and GJ—adm).

**Table 1 ijms-25-09815-t001:** Ka/Ks ratio of OsRS homologous genes.

Gene 1	Gene 2	Ka	Ks	Ka/Ks
*OsRS2*	*OsRS54*	0.137	0.283	0.484

**Table 2 ijms-25-09815-t002:** Comparison of genetic diversity of six OsRS genes in *Xian* between local and modern varieties.

Gene Name	*Xian* (Indica)							
	LANs		MVs		MVs—LANs		Statistical	Artificial Selection Effect
	*E_H_*	gcHapN	*E_H_*	gcHapN	*E_H_*	gcHapN		
*OsRS1*	0.127	26	0.229	28	0.102	2	**	up
*OsRS2*	0.090	6	0.103	7	0.013	1	ns	up
*OsRS3*	0.326	49	0.413	46	0.087	−3	**	up
*OsRS4*	0.263	62	0.344	44	0.081	−18	**	up
*OsRS5*	0.312	63	0.438	51	0.126	−12	**	up
*OsRS6*	0.214	19	0.229	11	0.016	−8	ns	up
Mean	0.250	42.4	0.329	35.1	0.079			

*E_H_* and gcHapN are Shannon’s fairness and the number of identified major gcHaps (≥1% varieties), respectively. ** indicate the statistical significance at levels of *p* < 0.01 and ns not significant, respectively based on Z tests.

**Table 3 ijms-25-09815-t003:** Comparison of genetic diversity of six OsRS genes in local and modern varieties in *Geng*.

Gene Name	*Geng* (Japonica)							
	LANs		MVs		MVs—LANs		Statistical	Artificial Selection Effect
	*E_H_*	gcHapN	*E_H_*	gcHapN	*E_H_*	gcHapN		
*OsRS1*	0.192	19	0.211	13	0.019	−6	ns	up
*OsRS2*	0.028	6	0.045	3	0.016	−3	ns	up
*OsRS3*	0.163	20	0.204	13	0.041	−7	ns	up
*OsRS4*	0.244	29	0.214	12	−0.029	−17	ns	down
*OsRS5*	0.257	27	0.300	18	0.043	−9	ns	up
*OsRS6*	0.129	8	0.205	8	0.076	0	*	up
Mean	0.171	18	0.210	12				

*E_H_* and gcHapN are Shannon’s fairness and the number of identified major gcHaps (≥1% varieties), respectively. * indicate the statistical significance at levels of *p* < 0.05 and ns not significant, respectively based on Z tests.

## Data Availability

Data are contained within the article and Appendix A.

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
