# Peer review of "Analysis of the Rice Raffinose Synthase (OsRS) Gene Family and Haplotype Diversity"

_ijms, 2024, doi:10.3390/ijms25189815_

Round 1

Reviewer 1 Report

Comments and Suggestions for Authors

In this manuscript, the authors present their analysis of raffinose synthase (OsRS) gene family in rice and the haplotype diversity of OsRS genes.

They employed genome information of Nipponbare rice cultivar and investigated the OsRS gene family for its physical and chemical properties, phylogenetic relationship, conserved domains, promoter cis-acting elements, and the functions of genes.

The authors identified six raffinose synthase (RS) genes on 6 different chromosomes comprising 3 major subfamilies in four major rice populations and studied them under several stress conditions such as drought, salt, heat and cold stress, and hormonal treatments (ABA, MeJA). The authors report that these genes play key roles in abiotic stress responses.

Moreover, the authors analyzed genetic diversity based on the coding sequence haplotype data (gcHap) in the OsRS gene family. The authors demonstrate that the main gcHaps of most OsRS loci had significant effects on key agronomic traits, and the frequency of these alleles varied significantly among different rice populations and subspecies.

Conclusively, it is evident that the findings may serve as basis for studying the RS gene family in other crops and will serve broader audiences.

Overall, the article is well written but from readers point of view I have one suggestion:

1. It is a general practice that authors make their R scripts available. So, I would encourage the authors to do so. 

Author Response

Comments1. It is a general practice that authors make their R scripts available. So, I would encourage the authors to do so.

Response1:Thank you for your insightful suggestion regarding the public availability of our R scripts. We have taken your advice into serious consideration and have carefully weighed the potential benefits and drawbacks.After thorough discussion, we have decided not to release the R scripts at this juncture. Our decision is grounded in the following considerations:

1)The R scripts in question are an integral part of our research on haplotype analysis, which is central to the findings presented in our manuscript.  However, these scripts are not yet ready for public disclosure due to their proprietary nature and the ongoing development of related methodologies.

2)We are currently engaged in follow-up studies that rely heavily on the methodologies encapsulated within these scripts.The premature release of our scripts could potentially compromise the novelty and impact of our future research findings.

We hope you understand our position and continue to support our research. Thank you for your understanding and continued support.

Reviewer 2 Report

Comments and Suggestions for Authors

The reason for this decision is:

This manuscript does not fulfill the standards established for the journal to be considered for publication.

This study performed OsRS gene identification and chromosome diversity analysis. However, the information in Nipponbare of rice was cis-acting (Fig. 2) The results of the Collinear relationship, as shown in Figure 3, were also presented. Chromosomal location association analysis, as shown in Figure 4, was also performed. In addition, OsRS gene expression analysis of six traits was performed, as shown in Figure 5. Figure 6 describes the expression of the OsRS series of six traits. The regrettable point in this study is that it is questionable how much scientific evidence is based on mechanical data. Therefore, it is questionable whether the transformation or gene editing of one related trait is I hereby inform you that this paper cannot be printed without conducting experiments. Therefore, this paper contains errors that cannot be corrected any further. Therefore, I think this paper is not suitable for this journal.

Author Response

Comments1. This study performed OsRS gene identification and chromosome diversity analysis. However, the information in Nipponbare of rice was cis-acting (Fig. 2) The results of the Collinear relationship, as shown in Figure 3, were also presented. Chromosomal location association analysis, as shown in Figure 4, was also performed. In addition, OsRS gene expression analysis of six traits was performed, as shown in Figure 5. Figure 6 describes the expression of the OsRS series of six traits. The regrettable point in this study is that it is questionable how much scientific evidence is based on mechanical data. Therefore, it is questionable whether the transformation or gene editing of one related trait is I hereby inform you that this paper cannot be printed without conducting experiments. Therefore, this paper contains errors that cannot be corrected any further. Therefore, I think this paper is not suitable for this journal.

Response1:We appreciate the feedback from the reviewers on our study and have taken particular note of your concerns regarding the scientific evidence and the reliability of the data. In response to the issues you've raised, we would like to clarify that our research already includes quantitative PCR (QPCR) experiments to ensure the quantitative accuracy of our preliminary analysis of the OsRS gene expression. However, we recognize the need for further research to deepen our understanding of the function of the OsRS gene. To this end, we plan to conduct more in-depth analyses, including but not limited to exploring the interactions between the OsRS gene and other genes, as well as performing additional gene editing and transformation experiments to verify its role in specific traits. We believe that these additional studies will provide a more solid foundation for our conclusions, and we look forward to submitting a more refined set of research results after completing this work.

Reviewer 3 Report

Comments and Suggestions for Authors

The title needs to be refined, moreover, the authors need to illustrate the sources of the rice germplasms used in this experiment. The abstract section needs to be adjusted to add the key findings

Comments on the Quality of English Language

The language is Ok, except for some minor grammatical errors, which can be easily polished by the authors.

Author Response

Comments1. The title needs to be refined, moreover, the authors need to illustrate the sources of the rice germplasms used in this experiment. The abstract section needs to be adjusted to add the key findings.

Response1:Thank you for your comments.The article title has been changed to“Analysis of rice raffinose synthase (OsRS) gene family and haplotype diversity”.

The “Material handling” supplement at Material Methods states that the rice used for the experiment is a Nipponbare seed retained in our laboratory, and the abstract and keyword parts of the manuscript have been adapted according to revisions.

Reviewer 4 Report

Comments and Suggestions for Authors

1.  The analysis primarily focuses on the gene family rather than specific gene identification, which may not perfectly align with the manuscript's title. Additionally, functional verification of gene mutation or overexpression is lacking in the manuscript.

2. What types of rice materials and components were utilized for the quantitative analysis of "4.6 Material handling" in the article?

3. The results showed that the expression of OsRS5 gene in leaves was significantly changed under high salt, drought, ABA and MeJA treatments, which played a key role in abiotic stress response. Do different haplotypes of the OsRS gene family exhibit tolerance or susceptibility phenotypes?

4. The article lacks standard writing conventions, such as the use of italics for statistical significance (e.g., *P < 0.05) and contains numerous formatting errors in the references.

Comments on the Quality of English Language

1.  The analysis primarily focuses on the gene family rather than specific gene identification, which may not perfectly align with the manuscript's title. Additionally, functional verification of gene mutation or overexpression is lacking in the manuscript.

2. What types of rice materials and components were utilized for the quantitative analysis of "4.6 Material handling" in the article?

3. The results showed that the expression of OsRS5 gene in leaves was significantly changed under high salt, drought, ABA and MeJA treatments, which played a key role in abiotic stress response. Do different haplotypes of the OsRS gene family exhibit tolerance or susceptibility phenotypes?

4. The article lacks standard writing conventions, such as the use of italics for statistical significance (e.g., *P < 0.05) and contains numerous formatting errors in the references.

Author Response

Comments1. The analysis primarily focuses on the gene family rather than specific gene identification, which may not perfectly align with the manuscript's title. Additionally, functional verification of gene mutation or overexpression is lacking in the manuscript.

Response1:Thank you for your comments.The article title has been changed to"Analysis of rice raffinose synthase (OsRS) gene family and haplotype diversity" .

Functional validation of gene deletion or overexpression will be performed later, and thank you for your valuable suggestions for this study.

Comments2. What types of rice materials and components were utilized for the quantitative analysis of "4.6 Material handling" in the article?

Response2:Thank you for your comments.The rice used for "4.6 Material handling" is Nipponbare and the leaves of rice are extracted, which has been supplemented in "4.6 Material handling".

Comments3.The results showed that the expression of OsRS5 gene in leaves was significantly changed under high salt, drought, ABA and MeJA treatments, which played a key role in abiotic stress response. Do different haplotypes of the OsRS gene family exhibit tolerance or susceptibility phenotypes?

Response3:Thank you for your comments.In this study, the different haplotype analysis of the OsRS gene family was not linked to the tolerance and susceptibility phenotype, but existed alone, so it is not possible to directly see the different tolerance or susceptibility phenotypes of OsRS genes.

Comments4.The article lacks standard writing conventions, such as the use of italics for statistical significance (e.g., *P < 0.05) and contains numerous formatting errors in the references.

Response4:Thank you for your comments.We greatly appreciate the reviewer's comments on the lack of standard writing conventions in our manuscript and the formatting errors in the references. We recognize the importance of these details in maintaining the rigor and professionalism of academic research. In response, we will take the following immediate actions: First, we will correct all statistical significance markers (e.g., *P < 0.05) to use italics, adhering to the standards of academic writing. Second, we will thoroughly review and correct the formatting of the references to ensure they comply with the journal's citation requirements. We thank you once again for your valuable feedback and look forward to submitting our revised manuscript after these improvements are made.

Round 2

Reviewer 4 Report

Comments and Suggestions for Authors

1. The statistical analysis in Figure 8 contains incorrect labels and should be carefully reviewed. For presenting the results of statistical analysis, please use either "ab" or "**" consistently throughout. The bar chart on the right displays significance levels as **** P < 0.0001, rendering it unnecessary to include *** P < 0.001, ** P < 0.01, * P < 0.05, and N.S.

2. Rephrase the figure legend to offer a comprehensive depiction of Figure 2a and b. Additionally, enhance the details in Figure 6 by including specific information on each stress treatment and statistical analysis methods employed.

Comments on the Quality of English Language

1. The statistical analysis in Figure 8 contains incorrect labels and should be carefully reviewed. For presenting the results of statistical analysis, please use either "ab" or "**" consistently throughout. The bar chart on the right displays significance levels as **** P < 0.0001, rendering it unnecessary to include *** P < 0.001, ** P < 0.01, * P < 0.05, and N.S.

2. Rephrase the figure legend to offer a comprehensive depiction of Figure 2a and b. Additionally, enhance the details in Figure 6 by including specific information on each stress treatment and statistical analysis methods employed.

Author Response

Comments1.The statistical analysis in Figure 8 contains incorrect labels and should be carefully reviewed. For presenting the results of statistical analysis, please use either "ab" or "**" consistently throughout. The bar chart on the right displays significance levels as **** P < 0.0001, rendering it unnecessary to include *** P < 0.001, ** P < 0.01, * P < 0.05, and N.S.

Response1:Thank you for your comments.Based on the revision suggestions, Figure 8 has been re-examined and modified, with the addition of labels for the boxplot. Additionally, the legend for Figure 8 has been revised to: "Haplotype networks of four cloned OsRS genes and four associated agronomic traits in 3KRG. The letters indicate differences between haplotypes assessed by two-factor ANOVA, where different letters on the boxplot indicate statistically significant differences at P < 0.05 based on Duncan's multirange test. The bar chart on the right shows the difference in frequency of major gcHaps between local varieties (LANs) and modern varieties (MVs) of Xian an and Geng. A Chi-square test was used to determine significant differences in the proportion of a gcHap between different populations **** P < 0.0001. It is noted that the boxplot and the bar chart on the right in Figure 8 use different statistical analysis methods for significance level analysis, hence "ab" or "**" are used to define their respective significance levels. For more operational steps, please refer to Zhang's Molecular Plant article, which is also cited in the manuscript, with the reference number 17."

Comments2.Rephrase the figure legend to offer a comprehensive depiction of Figure 2a and b. Additionally, enhance the details in Figure 6 by including specific information on each stress treatment and statistical analysis methods employed.

Response2:Thank you for your comments.Based on the suggestions, the legend for Figure 2 has been revised to: "Analysis of cis-acting elements of the OsRS gene. (a) Distribution and proportion of cis-acting elements in the promoter region of the OsRS gene, where different colors represent different proportions. (b) Heatmap Analysis of Cis-Acting Elements in the Promoter Region of OsRS Genes. In the heatmap, the numerical values represent the quantity of different cis-acting elements, with darker colors indicating higher quantities."The legend for Figure 6 has been revised to: "Analysis of expression levels of five genes of the OsRS family under different treatments. (a) 100 μmol/L ABA treatment. (b) 100μmol/L MeJA treatment. (c) 20% PEG6000 simulated drought stress. (d) 200 mmol/L NaCl simulated salt stress. (e) 42°C heat treatment. (f) 6°C cold treatment. Statistical analysis of the data was performed using WPS 2023 software, and analysis of variance was conducted using IBM SPSS Statistics 25 software. The significance level was defined as ****P<0.0001, ***P<0.001, **P<0.01, *P<0.05."